# Bionic Covert Underwater Acoustic Communication Based on Time–Frequency Contour of Bottlenose Dolphin Whistle

**DOI:** 10.3390/e24050720

**Published:** 2022-05-18

**Authors:** Lei Xie, Jiahui Zhu, Yuqing Jia, Huifang Chen

**Affiliations:** 1College of Information Science and Electronic Engineering, Zhejiang University, Hangzhou 310027, China; xiel@zju.edu.cn (L.X.); 22031078@zju.edu.cn (J.Z.); maven@zju.edu.cn (Y.J.); 2Zhejiang Provincial Key Laboratory of Information Processing, Communication and Networking, Hangzhou 310027, China; 3Zhoushan Ocean Research Center, Zhoushan 316021, China; 4State Key Laboratory of Fluid Power and Mechatronic Systems, Zhejiang University, Hangzhou 310027, China; 5The Engineering Research Center of Oceanic Sensing Technology and Equipment, Ministry of Education, Zhoushan 316021, China

**Keywords:** bionic covert communication, underwater acoustic communication, time–frequency contour, whistle, time-delay, frequency-shift

## Abstract

In order to meet the requirements of communication security and concealment, as well as to protect marine life, bionic covert communication has become a hot research topic for underwater acoustic communication (UAC). In this paper, we propose a bionic covert UAC (BC-UAC) method based on the time–frequency contour (TFC) of the bottlenose dolphin whistle, which can overcome the safety problem of traditional low signal–noise ratio (SNR) covert communication and make the detected communication signal be excluded as marine biological noise. In the proposed BC-UAC method, the TFC of the bottlenose dolphin whistle is segmented to improve the transmission rate. Two BC-UAC schemes based on the segmented TFC of the whistle, the BC-UAC scheme using the whistle signal with time-delay (BC-UAC-TD) and the BC-UAC scheme using the whistle signal with frequency-shift (BC-UAC-FS), are addressed. The original whistle signal is used as a synchronization signal. Moreover, the virtual time reversal mirror (VTRM) technique is adopted to equalize the channel for mitigating the multipath effect. The performance of the proposed BC-UAC method, in terms of the Pearson correlation coefficient (PCC) and bit error rate (BER), is evaluated under simulated and measured underwater channels. Numerical results show that the proposed BC-UAC method performs well on covertness and reliability. Furthermore, the covertness of the bionic modulated signal in BC-UAC-TD is better than that of BC-UAC-FS, although the reliability of BC-UAC-FS is better than that of BC-UAC-TD.

## 1. Introduction

Due to the serious attenuation of electromagnetic waves and the high dispersion of optical waves in water, sound waves have become the most effective method of underwater communication [1]. Nowadays, with the further development and utilization of ocean resources, underwater communication is playing an increasingly important role in military and civilian fields. However, the equipment is easily found by the enemy reconnaissance system due to the emission of acoustical signals during the communication process. Hence, underwater communication equipment requires concealment and security [2].

There are three types of covert underwater acoustic communication (UAC) method, namely low probability of detection (LPD) [3,4], low probability of recognition (LPR) [5], and low probability of interception (LPI) [6]. Moreover, LPD and LPI are two traditional covert UAC methods.

Since the detection probability is proportional to the signal-to-noise ratio (SNR), the LPD-based covert UAC method uses some technologies, such as orthogonal frequency division multiplexing (OFDM) [7], direct sequence spread spectrum (DSSS) [8], spatial diversity, frequency diversity and so on, to protect the marine ecological environment without exposing the communication equipment. However, as SNR decreases, the reliability, communication rate and communication distance of these LPD-based covert UAC schemes will obviously degrade.

On the other hand, the LPI-based covert UAC method is to encode the digital symbol in a high SNR [9]. This method will not only expose the communication equipment, but also affect the marine environment. However, this method performs well on reliability, communication rate and communication distance.

In the actual detection process, the passive sonar can still detect the LPD-based communication signal through a long-time integration. As the adversary approaches, the location of the communication equipment will be exposed. Moreover, since the signal transmitted by the LPD-based covert UAC method has obvious characteristics, the information is exposed, and the security is invalid once detected by the adversary. In addition, one finds that man-made noise can damage the sonar navigation system of cetaceans, which makes toothed whales unable to communicate, prey and identify directions, and even die in extreme cases [10,11,12]. Figure 1 illustrates the effect of human activities on toothed whales.

In recent years, the LPR-based covert UAC method, which uses bionic communication technology to improve communication distance and reliability, as well to protect the marine environment and eliminate the exposure of communication equipment, has become a hot research topic. The LPR-based covert UAC method is to imitate the sounds in the marine environment, such as the sounds of marine mammals, human activities and natural sounds, or artificially generated sound signals. For bionic covert UAC (BC-UAC), the communication signals are usually identified as marine environmental noise and filtered out in signal processing [13,14,15]. Hence, the BC-UAC method can ensure the security of communication and the concealment of communication equipment, as well as providing environmental friendliness.

Since the usage of sound to achieve covert UAC was first proposed by the Office of Naval Research in 2001 [16], many BC-UAC methods have been proposed [17,18,19,20,21,22,23,24,25,26,27,28,29,30,31].

Firstly, the characteristics of marine animal sound signals are used for modulation [17,18,19,20,21,22,23]. In [17], the time interval between dolphin clicks is used to modulate information bits, and the dolphin whistle is used as the synchronization signal. In [18], the dual-orthogonal modulation method is proposed based on the compressed sea lion click signal, as well as using the sea lion whistle signal as the synchronization signal. In addition, the concealment of communication is improved by passive time reversal mirror technology. In [19], a new frame hopping structure is proposed. The appropriate signal is selected from the original signals, according to the ambiguity function and correlation characteristic. In [20], a bionic camouflage communication system is designed, where the killer whale sound signal is used for communication address coding. In [21], a bionic Morse coding mimicking the humpback whale song is used in covert underwater communication. In [22], a bio-inspired covert UAC based on sperm whale clicks is proposed, where frequency-hopping modulation is used on different frequency band distributions of the sperm whale tick, and the encoded sequence is inserted into the corresponding position of the original signal.

Secondly, traditional signals and marine animal sound signals are used to synthesize the transmitting signals [23,24,25,26]. In [23], watermarking is embedded into the bottlenose dolphin signal, and the embedded information is extracted by an extended Prony method. In [24], a bionic communication scheme is proposed, where the minimum shift keying (MSK) modulated signal is copied to the dolphin whistle contour with continuous phase. To realize underwater target detection and covert communication, original clicks are combined with traditional sonar waveforms [25]. In [26], the framework of covert UAC is constructed, where the whistle is modulated by a special spectrogram pattern and demodulated by the correlation peak.

Thirdly, some researchers try to simulate the inherent acoustical signal to realize covert communication [27,28,29,30,31]. In [27], the navigation test of imitating dolphin whistles and clicks is completed with the pulse position modulation (PPM) method, but the corresponding demodulation scheme is not given. The generalized frequency modulated (GFM) signal is used to imitate the fundamental wave of a dolphin whistle, and information bits are mapped to the parameters of GFM biomimetic signals [28]. Moreover, based on imitated high frequency clicks, the time hopping code is used to modulate each bit, and the PPM is used to modulate the frame sequence [29]. In [30], a BC-UAC method based on camouflaging sea piling sounds is proposed, where the periodic PPM is adopted. In [31], a method of simulating ship-radiated noise with chaotic signals is proposed, and a BC-UAC scheme based on mimicking ship-radiated noise is presented.

Although many BC-UAC methods have been proposed in the past twenty years, the concealment and communication rate of these methods should be improved due to the complexity and variability of the underwater environment. Moreover, considering that the wide frequency band of clicks is not suitable for practical transducers, and the frequency band of the whistle signal is narrow and has good time–frequency characteristics, the whistle signal of cetaceans is selected for bionic covert communication. Furthermore, since the communication rate in the existing BC-UAC methods is generally low, the BC-UAC method with a higher communication rate is studied in this work. Hence, the problem of BC-UAC based on the time–frequency contour (TFC) of the bottlenose dolphin whistle is investigated in this paper.

In order to improve the concealment and effectiveness of the constructed bionic communication signal, a BC-UAC method based on the TFC of the bottlenose dolphin whistle is proposed. First, the TFC of the bottlenose dolphin whistle is extracted by the short-time Fourier transform (STFT) algorithm and smoothed by the Sage–Husa adaptive Kalman filter (SHAKF) algorithm. Second, the transmitted information bits are modulated to the TFC, where the TFC is processed as segments to improve the communication rate. Two BC-UAC schemes based on the segmented TFC of the bottlenose dolphin whistle, namely the BC-UAC scheme using the whistle signal with time-delay (BC-UAC-TD) and the BC-UAC scheme using the whistle signal with frequency-shift (BC-UAC-FS), are presented. In the proposed BC-UAC schemes, the information bits are represented by time-delay and frequency-shift to obtain the bionic modulated signal after modulation. Furthermore, an improved orthogonal matching pursuit (IOMP) algorithm is addressed to estimate the channel with unknown sparsity, and virtual time reversal mirror (VTRM) technology is used to perform channel equalization. The performance of the proposed BC-UAC method is evaluated with simulated and measured underwater channels, respectively. Numerical results show that the constructed bionic communication signals have good concealment. The two BC-UAC schemes have good performance in terms of covertness and reliability. In addition, the covertness of BC-UAC-TD is better than that of BC-UAC-FS, although the reliability of BC-UAC-TD is worse than that of BC-UAC-FS.

The remainder of this paper is organized as follows. In Section 2, we analyze the TFC characteristics of the bottlenose dolphin whistle. Two BC-UAC schemes based on the segmented TFC of the bottlenose dolphin whistle are presented in Section 3. In Section 4, the performance of the proposed BC-UAC method is evaluated with simulated and measured underwater channels. Finally, the conclusion of the paper is given in Section 5.

## 2. Feature Analysis of Bottlenose Dolphin Whistle

The dolphin is an advanced mammal in the ocean. The clicks of bottlenose dolphins are echolocation signals, with the frequency range of 10 Hz–150 kHz and the duration of 25 μs. After a long evolution, dolphin sonar has the function of communication. The whistles of bottlenose dolphins are signals for communication with the frequency range of 100 Hz–24 kHz and the duration of 1 s. The operating frequency of most underwater acoustic devices ranges from 1 kHz to 40 kHz [32], which overlaps with the frequency range of dolphin signals. Hence, it is feasible to use the sound of cetaceans for covert communication in the underwater environment. Since clicks are a kind of narrow pulse signal, it is difficult to make the transducer. Hence, the bottlenose dolphin whistle is selected as the reference signal for bionic covert communication.

The whistle is a frequency modulated signal whose frequency changes with time and has harmonic characteristics. Using the STFT algorithm to analyze the time–frequency characteristics, according to the time-varying trend of the TFC of a whistle signal, there are four categories of whistle, namely up-sweep whistle, down-sweep whistle, flat-sweep whistle and sinusoidal whistle, as illustrated in Figure 2. The original whistle contains noise and higher harmonics.

From Figure 2, one finds that the fundamental frequency energy of the whistle signal is usually the strongest. Therefore, a pure fundamental frequency signal is extracted as the whistle signal TFC. First, the time–frequency spectrogram of the whistle signal is obtained by the STFT algorithm. In addition, the frequency with the largest energy in each segment is taken as the frequency at the beginning of the segment. The fundamental frequency TFC can be expressed as:(1)$$f[n]=\underset{f}{\arg\max}\;|X[n,f]|,n=1{,}^{}2{,}^{}\cdots {,}^{}T_{\mathrm{whistle}}\times f_{s}$$
where *X*[*n*, *f*] is a two-dimensional function of sampling time *n* and frequency *f* is the result of the STFT algorithm; *f_s_* is the sampling frequency; and *T*_whistle_ is the duration of the whistle signal.

The waveform of a bottlenose dolphin whistle and the corresponding TFC, smoothed by the SHAKF algorithm, are shown in Figure 3a,b, respectively. From Figure 3, we observe that it is a down-sweep whistle, where the frequency band ranges from 11.8 kHz to 6.5 kHz, and the duration is about 0.53 s. The BC-UAC method proposed in this work is based on the segmented TFC of the bottlenose dolphin whistle shown in Figure 3.

The normalized autocorrelation of the whistle shown in Figure 3a is illustrated in Figure 4. From Figure 4, we observe that the autocorrelation peak of the bottlenose dolphin whistle is sharp and prominent, which means that it has a good autocorrelation characteristic. Hence, the BC-UAC method proposed in this paper uses the original whistle as the synchronization signal.

## 3. Proposed BC-UAC Method

In this section, we propose a BC-UAC method based on the TFC of the bottlenose dolphin whistle.

### 3.1. The Frame Structure

Figure 5 illustrates the frame structure of BC-UAC based on the bottlenose dolphin whistle, which is composed of the synchronization signal, the guard interval and the bionic modulated information bits.

Due to its good autocorrelation characteristic, the original whistle signal is used as the synchronization signal. By performing the correlation between the received whistle signal and the original whistle signal stored locally, the location of the correlation peak is obtained to realize the synchronization of the frame and the recognition of the whistle signal.

A blank space is used as the guard interval to prevent the interference between the synchronization signal and the bionic modulated information bits caused by the multipath effect of the underwater channel.

The bionic modulated information bits generate a synthetic whistle modulated by time-delay or frequency-shift.

### 3.2. The BC-UAC System Model

Figure 6 shows the block diagram of the BC-UAC system based on the TFC of the bottlenose dolphin whistle, which consists of the transmission module, the reception module and the underwater channel. In the proposed BC-UAC method, the TFC of the bottlenose dolphin whistle is modulated by segments in order that each whistle signal can carry more information bits.

In the transmission module, the TFC of the bottlenose dolphin whistle shown in Figure 3a, which is extracted by the STFT algorithm and smoothed by the SHAKF algorithm, is segmented. The precoded source sequence is modulated to the segmented TFC of the bottlenose dolphin whistle to generate the bionic modulated signal. Then, the frame of BC-UAC is constructed with the original whistle signal, the blank interval and the generated bionic modulated signal.

Suffering from the effect of the underwater channel, the signal reaches the receiver.

In the reception module, the correlation between the received synchronization signal and the original whistle signal stored locally is calculated to locate the beginning of the received bionic modulated signal. Then, the received bionic modulated signal is equalized to reduce the multipath effect of the underwater channel. The bionic modulated signal is demodulated and decoded to output the received information bits.

### 3.3. The Transmission Scheme Based on Segmented TFC

The whistle is an important component of the dolphin’s communication. The whistle has the characteristics of continuous time and concentrated frequency distribution, and its TFC acoustic characteristic is obvious. Hence, we chose to modulate the TFC of the whistle to improve the concealment of the communication signal. The basic idea, based on segmented TFC modulation, is to modulate different information on each segment of the TFC of the whistle signal.

First, let ***d*** = {*d*_1_, *d*_2_, …, *d_P_*} be the source sequence, where *P* is the length of ***d***. According to the modulation order ***M***, the source sequence is precoded to become an ***M***-ary coded sequence, ***b*** = {*b*_1_, *b*_2_, …, *b_K_*}, where *K* is the length of ***b***, $K=[P/\log_{2}M]$, and [·] denotes the operation of rounding up to an integer.

At the same time, the TFC of the bottlenose dolphin whistle is extracted by Formula (1) and smoothed by the SHAKF algorithm. Then, the TFC is evenly divided into *K* segments. *T*_sym_ is the duration of a symbol and *T*_sym_ = *T*_whistle_/*K*.

In this work, two BC-UAC schemes, namely BC-UAC-TD and BC-UAC-FS, are designed to modulate the coded sequence ***b*** on the TFC segments.

#### 3.3.1. For BC-UAC-TD

In BC-UAC-TD, the TFC segment is divided into *M* subsegments, and each subsegment is modulated according to its time-delay and shifted by Δ*f* as a marker. The duration of a subsegment is *T*_0_ = *T*_sym_/*M*.

The time-delay of the *k*-th symbol is *τ_k_* = *b_k_**T*_0_, where *b_k_* is the *k*-th element of the coded sequence, and *k* = 1, 2, …, *K*. The frequency offset to characterize the time-delay in BC-UAC-TD is illustrated in Figure 7.

Hence, the modulated frequency of symbol *k*, *f′*_whistle,*k*_[*n*], is:(2)$${f^{\prime }}_{\mathrm{whistle},k}[n]=\{\begin{array}{cc} f_{\mathrm{whistle},k}[n]+\Delta f, & \;(k-1)T_{\mathrm{sym}}+\tau_{k}<n\leq (k-1)T_{\mathrm{sym}}+\tau_{k}+T_{0}\\ f_{\mathrm{whistle},k}[n]\;, & \mathrm{others} \end{array}$$
where, *f*_whistle,*k*_[*n*] is the initial frequency of whistle in (1), and Δ*f* is the frequency offset to represent the time-delay.

The whistle signal can be viewed as the weighted superposition of a series of cosine signals with envelope and frequency changes [33]. The bionic modulated signal in BC-UAC-TD can be expressed as:(3)$$s_{\mathrm{TD}}[n]=a[n]\cos\left(2\pi \sum_{i=1}^{n}\frac{{f^{\prime }}_{\mathrm{whistle}}[i]}{f_{s}}\right),n=1{,}^{}2{,}^{}\cdots {,}^{}T_{\mathrm{whistle}}\times f_{s}$$
where *a*[*n*] is the envelope of the whistle signal at sampling time *n*.

#### 3.3.2. For BC-UAC-FS

Since the division of TFC segments in BC-UAC-TD results in energy loss, the different frequency-shift is used to modulate the coded sequence in BC-UAC-FS. The frequency-shift in BC-UAC-FS is illustrated in Figure 8.

For symbol *k*, the frequency-shift is:(4)$$\Delta f_{k}=b_{k}\Delta f_{0},k=1,2,\cdots ,K$$
where ∆*f*_0_ is the unit frequency offset.

The modulated frequency of symbol *k*, *f″*_whistle,*k*_[*n*], is:(5)$${f^{\prime \prime }}_{\mathrm{whistle},k}[n]=f_{\mathrm{whistle},k}[n]+b_{k}\Delta f_{0},(k-1)T_{\mathrm{sym}}<n\leq kT_{\mathrm{sym}}$$

The bionic modulated signal in BC-UAC-FS can be expressed as:(6)$$s_{\mathrm{FS}}[n]=a[n]\cos\left(2\pi \sum_{i=1}^{n}\frac{{f^{\prime \prime }}_{\mathrm{whistle}}[i]}{f_{s}}\right),n=1{,}^{}2{,}^{}\cdots {,}^{}T_{\mathrm{whistle}}\times f_{s}$$

### 3.4. The Reception Scheme Based on Segmented TFC

At the receiver, the correlation operation between the original whistle signal stored locally and the received synchronization signal is performed to synchronize the received signal.

#### 3.4.1. Channel Estimation and Equalization

Due to the large delay spread and obvious Doppler effect of the underwater communication channel, channel estimation and equalization technology should be adopted to improve the transmission performance. If the Doppler frequency offset is eliminated after variable sampling, the impulse response of the time-invariant underwater acoustic multipath channel can be expressed as:(7)$$h(t)=\sum_{l=1}^{L}\alpha_{l}\delta (t-\tau_{l})\;$$
where *L* is the number of multipaths, and *α_l_* and *τ_l_* are the amplitude decline and time-delay of the *l*-th path, respectively.

At the receiver, the multipath signals with different attenuations and delays are superimposed, which will introduce inter-symbol interference. In order to alleviate the influence of the multipath effect, the VTRM technique [34] combined with an IOMP algorithm is adopted for the channel equalization. The workflow of the channel estimation and equalization in the proposed BC-UAC method is illustrated in Figure 9, where the IOMP algorithm is used for the channel estimation, and the estimated results are used for the channel equalization.

The IOMP algorithm is an improvement of OMP. Considering that the sparsity of the actual underwater channel is unknown, the number of iterations of the OMP algorithm depends on the effective length of the channel, which results in over-calculation. In the IOMP algorithm, a weak selection factor is introduced to obtain several matching vectors in each iteration. The residual threshold can be set as the ending condition of iteration. Hence, the IOMP algorithm speeds up the iteration speed, but the result of each iteration is suboptimal, and the channel estimation is slightly less accurate than that of the OMP algorithm.

Let $\hat{h}\left[n\right]$ denote the channel impulse response (CIR), estimated by the IOMP algorithm at sampling time *n*, $\hat{h}\left[-n\right]$ be the CIR after time-reversal, and *r*[*n*] be the received bionic modulated signal. After the convolution of *r*[*n*] and $\hat{h}\left[-n\right]$, the virtual received bionic modulated signal *r’*[*n*] is obtained. Hence, the process of the VTRM can be expressed as:(8)$$\begin{array}{ll} r^{\prime }[n] & =r[n]*\hat{h}[-n]\\ & =\left(s[n]*h[n]+\varsigma [n]\right)*\hat{h}[-n]\\ & =s[n]*h^{\prime }[n]+\varsigma [n]*\hat{h}[-n] \end{array}$$
where *h*[*n*] is the real underwater channel, $h^{\prime }\left[n\right]=h\left[n\right]*\hat{h}\left[-n\right]$ is the virtual time-reversal channel, ς[*n*] is the additive noise in the underwater channel, and *s*[*n*] is *s*_TD_[*n*] in (3) or *s*_FS_[*n*] in (6). *h’*[*n*] can be regarded as the equivalent channel through which the bionic communication signals pass. When $\hat{h}\left[n\right]$ is close to *h*[*n*], the correlation peak amplitude is much higher than the side lobe. It is equivalent to the multipath signal energy superimposed on the main path signal. Hence, the VTRM can suppress multipath interference and achieve equalization.

#### 3.4.2. For BC-UAC-TD

The received bionic modulated signal is evenly divided into *K* symbols after channel estimation and equalization. The received symbol *k* in BC-UAC-TD, *r′*_TD,*k*_[*n*], is:(9)$${r^{\prime }}_{\mathrm{TD},k}[n]=a^{\prime }[n]\cos\left(2\pi \sum_{i=(k-1)T_{\mathrm{sym}}+1}^{n}\frac{{f^{\prime }}_{\mathrm{whistle},k}[i]}{f_{s}}\right),n=1{,}^{}2{,}^{}\cdots {,}^{}T_{\mathrm{whistle}}\times f_{s}$$
where *a′*[*n*] is the envelope of the received bionic modulated signal, and $a^{\prime }\left[n\right]=\sum_{l=1}^{L}\alpha_{l}$.

The original whistle segment *k* is:(10)$$s_{\mathrm{TD}}{}_{,k}[n]=a[n]\cos\left(2\pi \sum_{i=(k-1)T_{\mathrm{sym}}+1}^{n}\frac{f_{\mathrm{whistle},k}[i]}{f_{s}}\right),n=1{,}^{}2{,}^{}\cdots {,}^{}T_{\mathrm{whistle}}\times f_{s}$$

By multiplying *r′*_TD,*k*_[*n*] and *s*_TD,*k*_[*n*], we have the coherent signal of symbol *k* in BC-UAC-TD, *g*_TD,*k*_[*n*], as:(11)$$\begin{array}{l} g_{\mathrm{TD},k}[n]=s_{\mathrm{TD},k}[n]{r^{\prime }}_{\mathrm{TD},k}[n]\\ =\frac{1}{2}a[n]a^{\prime }[n][\cos\left(2\pi \sum_{i=(k-1)T_{\mathrm{sym}}+1}^{n}\frac{{f^{\prime }}_{\mathrm{whistle},k}[i]+f_{\mathrm{whistle},k}[i]}{f_{s}}\right)+\cos\left(2\pi \sum_{i=(k-1)T_{\mathrm{sym}}+1}^{n}\frac{{f^{\prime }}_{\mathrm{whistle},k}[i]-f_{\mathrm{whistle},k}[i]}{f_{s}}\right)],\\ (k-1)T_{\mathrm{sym}}+\tau_{k}<n\leq (k-1)T_{\mathrm{sym}}+\tau_{k}+T_{0} \end{array}$$

For *g*_TD,*k*_[*n*], a low frequency signal component and a high frequency signal component can be obtained at the corresponding time-delay position. The low frequency signal is filtered by a low-pass filter, that is,
(12)$${g^{\prime }}_{\mathrm{TD},k}[n]=\frac{1}{2}a[n]a^{\prime }[n]\cos\left(2\pi \sum_{i=(k-1)T_{\mathrm{sym}}+1}^{n}\frac{{f^{\prime }}_{\mathrm{whistle},k}[i]-f_{\mathrm{whistle},k}[i]}{f_{s}}\right),(k-1)T_{\mathrm{sym}}+\tau_{k}<n\leq (k-1)T_{\mathrm{sym}}+\tau_{k}+T_{0}$$

According to (2), *f′*_whistle,*k*_[*i*] − *f*_whistle,*k*_[*i*] = ∆*f*. The coded data *b_k_* is demodulated according to time-delay *τ_k_*, corresponding to the maximum energy of frequency offset ∆*f* in symbol *k*. That is,
(13)$${\hat{b}}_{k}=\frac{\tau_{k}}{T_{0}},\;k=1,\;2,\;\ldots ,\;K$$

The original whistle is not a constant envelope signal, and its time-domain envelope changes with time. Hence, the energy of each TFC segment for a symbol is not same in the modulation, and the energy difference between two symbols may be large.

When the bionic modulated signal reaches the receiver through the underwater channel, multipath propagation often occurs. The multipath may cause serious inter-code interference between the symbols with high energy and low energy. Hence, in BC-UAC-TD, the influence of the multipath effect is more serious than that of a traditional UAC method.

In BC-UAC-TD, an energy compensation (EC) technique is adopted. The energy of each time-delay subsegment is {*E*_1_, *E*_2_, …, *E_M_*}. The compensation coefficient can be calculated as:(14)$$c_{m}=\frac{E_{\max}}{E_{m}},m=1,2,\cdots ,M$$
where *E*_max_ = max{*E*_1_, *E*_2_, …, *E_M_*}. The energy of each time-delay subsegment *m* is multiplied by the corresponding EC coefficient.

#### 3.4.3. For BC-UAC-FS

In BC-UAC-FS, the demodulation process is the same as that of BC-UAC-TD. However, in BC-UAC-FS, the symbol is not to be divided, and (*k* − 1)*T*_sym_ < *n* ≤ *kT*_sym_.

The coded data *b_k_* is demodulated according to frequency offset ∆*f_k_* in symbol *k*. That is,
(15)$${{\hat{b}}^{\prime }}_{k}=\frac{\Delta f_{k}}{\Delta f_{0}},\;k=1,\;2,\;\ldots ,\;K$$

The demodulated coded data are further decoded to obtain the received information bits, ${\hat{d}}^{\prime }$.

## 4. Simulation Results and Discussions

In this section, the performance of the proposed BC-UAC method is evaluated with simulated and measured underwater channels.

The Pearson correlation coefficient (PCC) is used to measure the concealment of the biomimetic modulated signal. The BER is used to measure the reliability of the proposed BC-UAC method.

The PCC is widely used to compare the similarity between two data sets [35]. Let the sets of the TFC sampling points of the original whistle signal before modulation and after modulation be $\left\{f_{\mathrm{whistle}}^{\left(1\right)},f_{\mathrm{whistle}}^{\left(2\right)},\cdots ,f_{\mathrm{whistle}}^{\left(I\right)}\right\}$ and $\left\{f_{\mathrm{whistle}}^{\prime \left(1\right)},f_{\mathrm{whistle}}^{\prime \left(2\right)},\cdots ,f_{\mathrm{whistle}}^{\prime \left(I\right)}\right\}$, where *I* is the number of sampling points in the set. The PCC of $\left\{f_{\mathrm{whistle}}^{\left(1\right)},f_{\mathrm{whistle}}^{\left(2\right)},\cdots ,f_{\mathrm{whistle}}^{\left(I\right)}\right\}$ and $\left\{f_{\mathrm{whistle}}^{\prime \left(1\right)},f_{\mathrm{whistle}}^{\prime \left(2\right)},\cdots ,f_{\mathrm{whistle}}^{\prime \left(I\right)}\right\}$ can be calculated as
(16)$$PCC=\frac{\sum_{i=1}^{I}\left(f_{\mathrm{whistle}}^{\left(i\right)}-{\bar{f}}_{\mathrm{whistle}}^{}\right)\left({f^{\prime }}_{\mathrm{whistle}}^{\left(i\right)}-{{\bar{f}}^{\prime }}_{\mathrm{whistle}}^{}\right)}{\sqrt{\sum_{i=1}^{I}{\left(f_{\mathrm{whistle}}^{\left(i\right)}-{\bar{f}}_{\mathrm{whistle}}^{}\right)}^{2}\sum_{i=1}^{I}{\left({f^{\prime }}_{\mathrm{whistle}}^{}-{{\bar{f}}^{\prime }}_{\mathrm{whistle}}^{}\right)}^{2}}}$$
where ${\bar{f}}_{\mathrm{whistle}}$ and ${\bar{f}}_{\mathrm{whistle}}^{\prime }$ are the average frequencies of the original whistle signal at each sampling time before and after modulation, respectively. When the value of the PCC approaches 1, it indicates high similarity between the bionic modulated signal and the original whistle signal. The larger the value of the PCC, the better the bionics of the modulated signal and the stronger the concealment.

### 4.1. PCC of Biomimetic Modulated Signal

In this subsection, the PCC of the biomimetic modulated signal generated by the proposed BC-UAC method is evaluated.

The impact of the symbol length (*T*_sym_) on the PCC is given in Table 1, where *M* = 4, ∆*f* = 200 Hz in BC-UAC-TD and ∆*f*_0_ = 200 Hz in BC-UAC-FS. From Table 1, we observe that the PCCs of BC-UAC-TD and BC-UAC-FS are greater than 0.98. Moreover, as *T*_sym_ decreases, the PCC of BC-UAC-TD remains unchanged, while the PCC of BC-UAC-FS increases gradually. The reason for this phenomenon is that the division of symbol length has an influence on BC-UAC-FS.

The impact of the frequency offset on the PCC is given in Table 2, where frequency-shift (∆*f*) is used in BC-UAC-TD, unit frequency-shift (∆*f*_0_) is used in BC-UAC-FS, *M* = 4, and *T*_sym_ = 0.066 s. From Table 2, we observe that as ∆*f* increases, the PCC of BC-UAC-TD changes a little and is greater than 0.99, which is very similar to the original whistle signal. For BC-UAC-FS, as ∆*f*_0_ increases, the PCC obviously decreases.

The impact of the modulation order (*M*) on the PCC is given in Table 3, where *T*_sym_ = 0.066 s, ∆*f* = 200 Hz in BC-UAC-TD and ∆*f*_0_ = 200 Hz in BC-UAC-FS. From Table 3, we observe that as *M* increases, the PCC of BC-UAC-TD increases a little, while the PCC of BC-UAC-FS decreases gradually, and the change is relatively large. Moreover, with the increase in *M*, the number of information bits in each symbol increases and the corresponding communication rate increases.

Hence, under the same conditions, the concealment of the biomimetic modulated signal in BC-UAC-TD is better than that in BC-UAC-FS.

### 4.2. BER under Simulated Underwater Channel

In this subsection, the BER of BC-UAC-TD and BC-UAC-FS is verified with the BELLHOP simulated channel model.

The parameter settings of the BELLHOP channel are listed in Table 4, and the corresponding normalized CIR is illustrated in Figure 10. In the simulated shallow water channel, the largest time-delay of the multipath is about 0.5 s, and its amplitude is about 0.15.

Figure 11 shows the impact of the symbol length (*T*_sym_) on the BER of BC-UAC-TD and BC-UAC-FS, where *M* = 4, ∆*f* = 200 Hz in BC-UAC-TD and ∆*f*_0_ = 200 Hz in BC-UAC-FS. From Figure 11, we observe that the BER of the proposed BC-UAC method decreases along with the increase in *T*_sym_. This is because the larger the *T*_sym_, the more energy each symbol has. As the SNR increases, the performance of the channel estimation and equalization module is improved. From Figure 11a, we observe that the error platform of the BER is eliminated by the EC technique in BC-UAC-TD. Comparing Figure 11a with Figure 11b, one finds that the performance in terms of the BER of BC-UAC-FS is better than that of BC-UAC-TD under the same conditions. The reason for this phenomenon is that the signal energy utilization rate of BC-UAC-FS is greater than that of BC-UAC-TD. For BC-UAC-TD, only part of the energy in the corresponding delayed segment is used. Moreover, with a given *T*_sym_, as *M* increases, the difference of energy utilization between BC-UAC-TD and BC-UAC-FS becomes greater. Using the VTRM in (8), the multipath effect is mitigated effectively, and the energy of the main path signal increases.

Figure 12 shows the impact of the frequency offset on the BER of BC-UAC-TD (∆*f*) and BC-UAC-FS (∆*f*_0_), where *M* = 4 and *T*_sym_ = 0.066 s. From Figure 12, we observe that the BER of the proposed BC-UAC method decreases gradually along with the increase in the frequency offset. This is because the larger the frequency offset, the higher the frequency resolution. Moreover, from Figure 12, we observe that the BER can be decreased using the EC technique in BC-UAC-TD, and the VTRM technique for channel equalization in the proposed BC-UAC method. In addition, with a given *T*_sym_ and different ∆*f* in BC-UAC-TD or ∆*f*_0_ in BC-UAC-FS, the performance in terms of the BER of BC-UAC-FS is better than that of BC-UAC-TD.

Figure 13 shows the impact of the modulation order (*M*) on the BER of the proposed BC-UAC method, where *T*_sym_ = 0.066 s, ∆*f* = 200 Hz in BC-UAC-TD and ∆*f*_0_ = 200 Hz in the BC-UAC-FS. From Figure 13, we observe that as *M* increases, the BER of the proposed BC-UAC method increases. The BER of BC-UAC-TD increases more than that of BC-UAC-FS. This is because the delay segment of a symbol in BC-UAC-TD becomes smaller as *M* increases, which results in a smaller signal energy used for modulation information. However, as *M* increases, the frequency band used for modulation information in BC-UAC-FS becomes larger, and the signal energy remains unchanged. Since the frequency resolution remains unchanged, the influence of *M* on the BER of BC-UAC-FS is relatively small.

### 4.3. BER under Measured Underwater Channel

In this subsection, the performance in terms of the BER of BC-UAC-TD and BC-UAC-FS is verified under the measured underwater channel.

The experiment was carried out in the Zhoushan sea area in October 2020. In the experiment, the transmitter was fixed on the ship and the receiver was fixed on the wharf. The depth of the seawater was about 60 m, the depths of the transmitter and the receiver were about 15 m and 5 m, respectively.

The impulse response of the measured underwater channel was estimated using the IOMP algorithm, as shown in Figure 14. From Figure 14, one finds that there are about 11 paths in the measured underwater channel, where the largest time-delay of the multipath is about 29.7 ms, and its corresponding amplitude is about 0.12.

The impact of the symbol length (*T*_sym_), the frequency offset and the modulation order (*M*) on the performance of the proposed BC-UAC method in terms of the BER under the measured underwater channel is shown in Figure 15, Figure 16 and Figure 17, respectively. The other simulation parameters are the same as in Table 4.

From Figure 15 and Figure 16, we observe that the performance of the proposed BC-UAC method in terms of the BER can be improved by increasing *T*_sym_, ∆*f* in BC-UAC-TD and ∆*f*_0_ in BC-UAC-FS. Moreover, the influence of *T*_sym_ is relatively small, while the influence of the frequency offset is relatively large.

From Figure 17, we observe that as the performance of the proposed BC-UAC method, in terms of the BER, decreases as *M* increases. However, the performance is improved using the VTRM equalization module at the receiver. Moreover, the influence of *M* on the performance of BC-UAC-TD is larger than that of BC-UAC-FS.

Comparing Figure 12, Figure 13 and Figure 14 with Figure 15, Figure 16 and Figure 17, shows the results obtained under the simulated underwater channel are in accordance with those under the measured underwater channel.

## 5. Conclusions

In this paper, we investigated the BC-UAC problem, and proposed a BC-UAC method based on the TFC of the bottlenose dolphin whistle. In the proposed BC-UAC method, the TFC is evenly divided into several segments to increase the communication rate, and the information bits are modulated by time-delay in BC-UAC-TD and frequency-shift in BC-UAC-FS, for each symbol. Moreover, since the whistle is not a constant envelope signal, the performance of the proposed BC-UAC method is more seriously affected by the multipath effect than the traditional UAC. In order to mitigate the multipath effect, the energy compensation technique, channel estimation with the IOMP algorithm and channel equalization with the VTRM technique are considered in the proposed BC-UAC method. Finally, the performance of the proposed BC-UAC method, in terms of the PCC and the BER, is evaluated under simulated and measured underwater channels. Numerical results show that the concealment of the biomimetic modulated signal behaves well in the proposed BC-UAC method. Moreover, BC-UAC-TD and BC-UAC-FS have good performance in terms of the BER. In addition, comparing BC-UAC-TD with BC-UAC-FS, the covertness of the biomimetic modulated signal in BC-UAC-TD is better than that of BC-UAC-FS, although the reliability of BC-UAC-TD is worse than that of BC-UAC-FS.

In future research, we will study the BC-UAC problem under the time-variant underwater acoustic multipath channel and improve the existing receiving technology. Moreover, we will verify the proposed BC-UAC method in the sea experiment. Furthermore, we will investigate the integrated communication and localization system based on the proposed BC-UAC method.

## Figures and Tables

**Figure 1 entropy-24-00720-f001:**
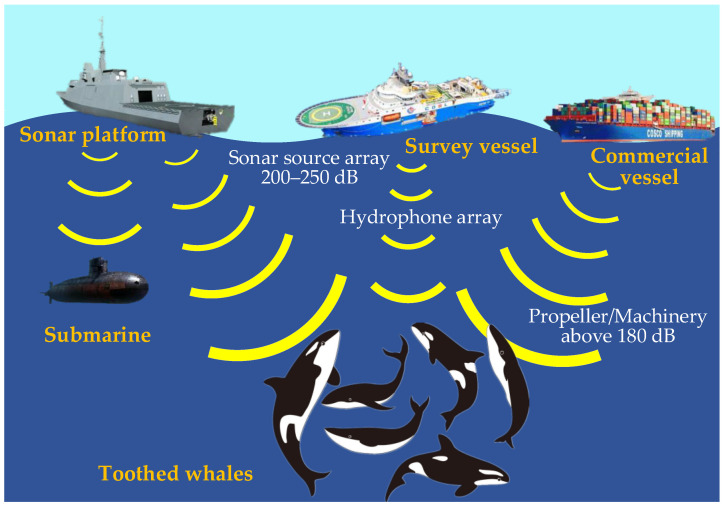
The effect of human activities on toothed whales.

**Figure 2 entropy-24-00720-f002:**
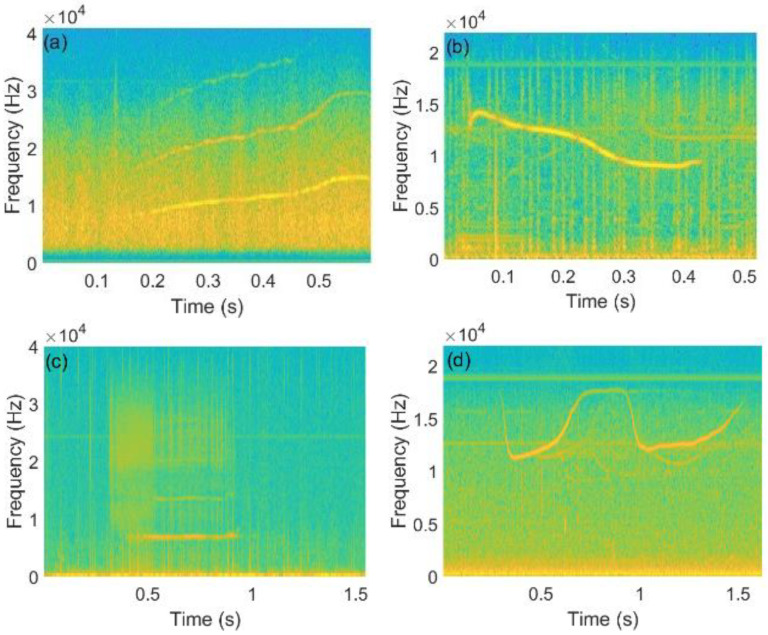
Four categories of whistle. (**a**) up-sweep whistle. (**b**) down-sweep whistle. (**c**) flat-sweep whistle. (**d**) sinusoidal whistle.

**Figure 3 entropy-24-00720-f003:**
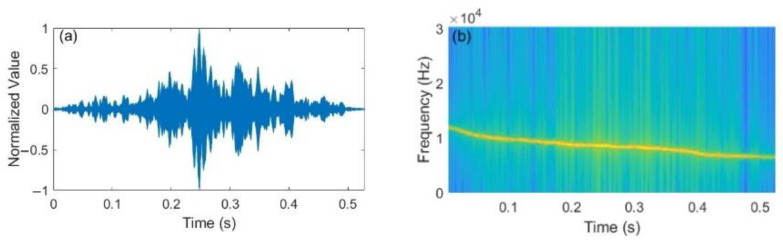
A bottlenose dolphin whistle. (**a**) waveform. (**b**) TFC.

**Figure 4 entropy-24-00720-f004:**
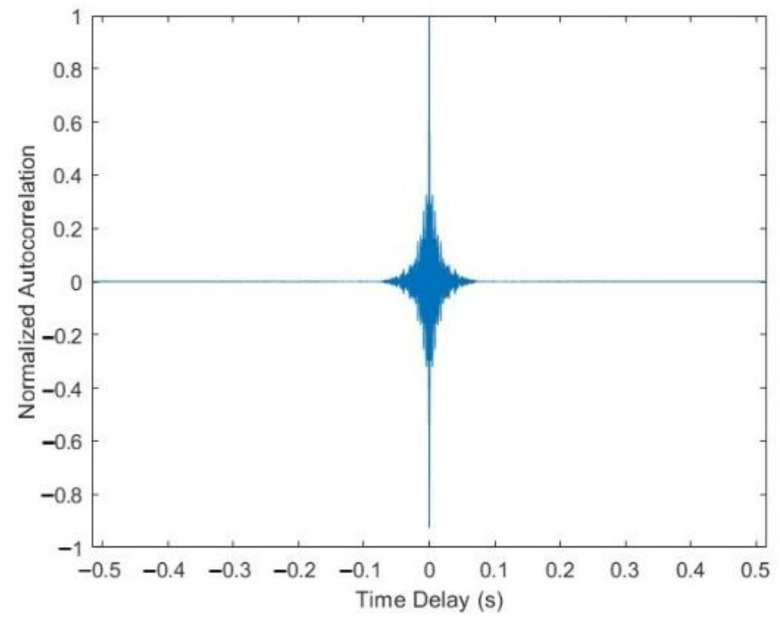
The normalized autocorrelation of the whistle shown in Figure 3a.

**Figure 5 entropy-24-00720-f005:**
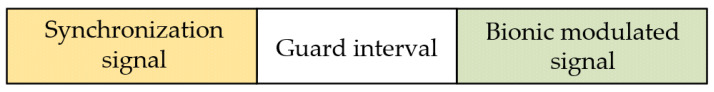
The frame structure of covert communication.

**Figure 6 entropy-24-00720-f006:**
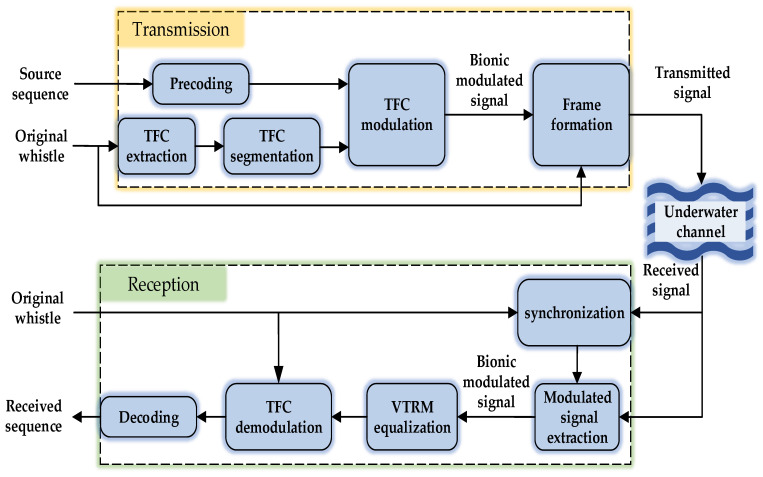
The model of the proposed BC-UAC system.

**Figure 7 entropy-24-00720-f007:**
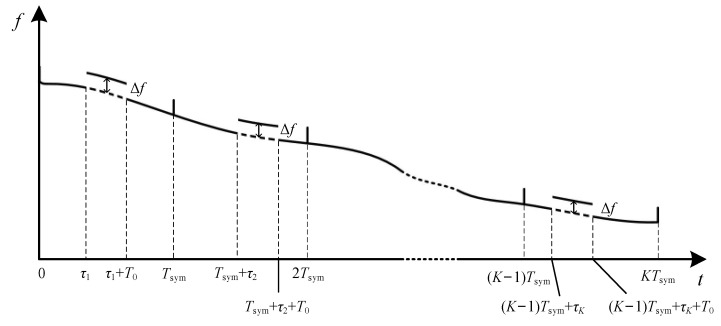
The frequency offset to characterize the time-delay in BC-UAC-TD.

**Figure 8 entropy-24-00720-f008:**
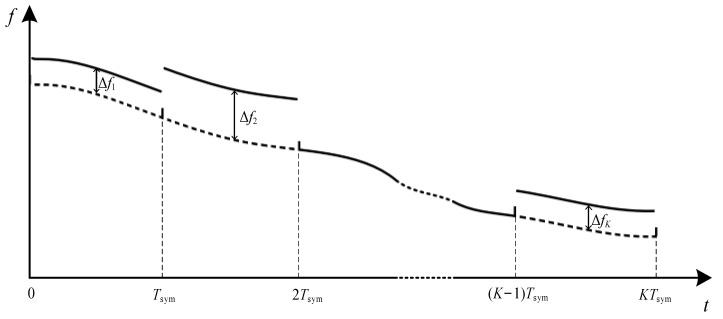
The frequency-shift in BC-UAC-FS.

**Figure 9 entropy-24-00720-f009:**
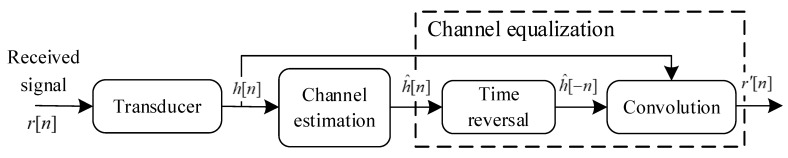
The workflow of the channel estimation and equalization in the BC-UAC method.

**Figure 10 entropy-24-00720-f010:**
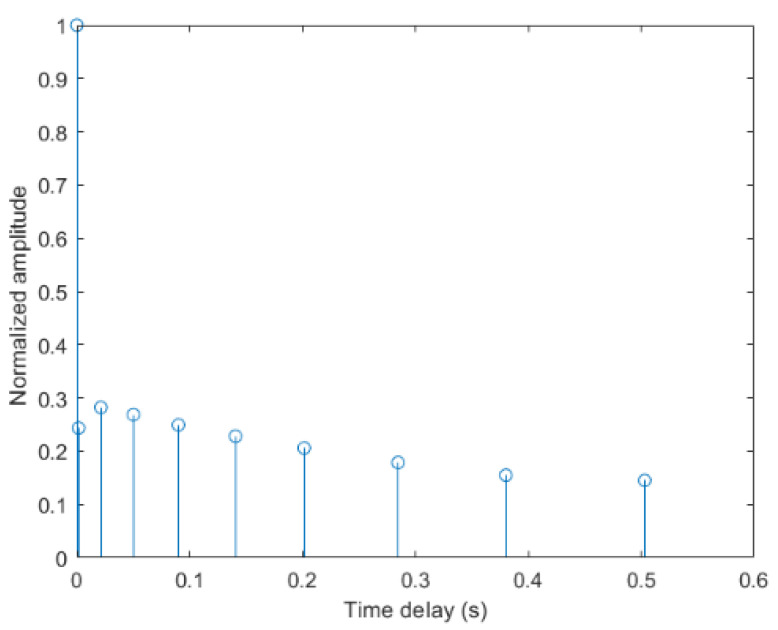
The impulse response of the BELLHOP channel.

**Figure 11 entropy-24-00720-f011:**
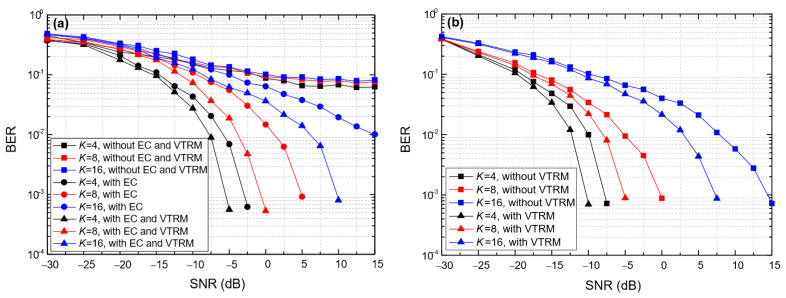
The impact of *T*_sym_ on the BER of the BC-UAC method under the simulated channel. (**a**) BC-UAC-TD. (**b**) BC-UAC-FS.

**Figure 12 entropy-24-00720-f012:**
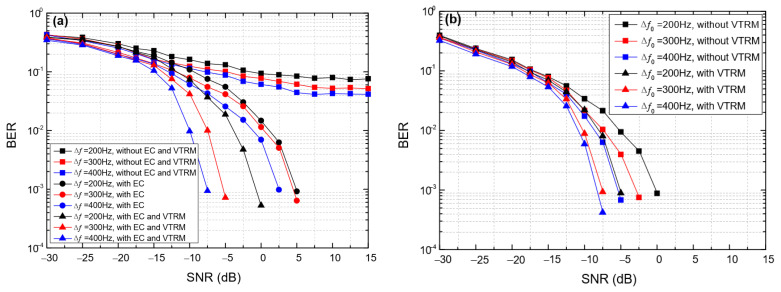
The impact of the frequency offset on the BER of the BC-UAC method under the simulated channel. (**a**) BC-UAC-TD (∆*f*); (**b**) BC-UAC-FS (∆*f*_0_).

**Figure 13 entropy-24-00720-f013:**
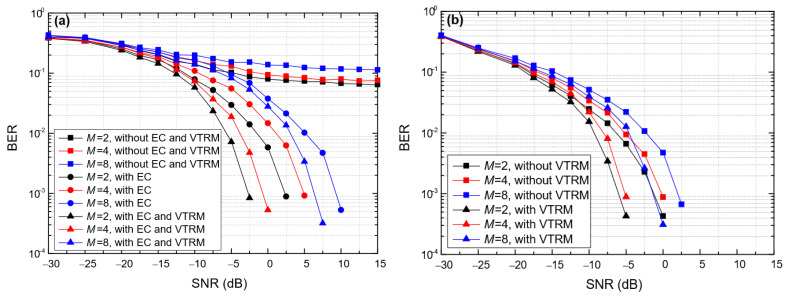
The impact of *M* on the BER of the proposed BC-UAC method under the simulated channel. (**a**) BC-UAC-TD; (**b**) BC-UAC-FS.

**Figure 14 entropy-24-00720-f014:**
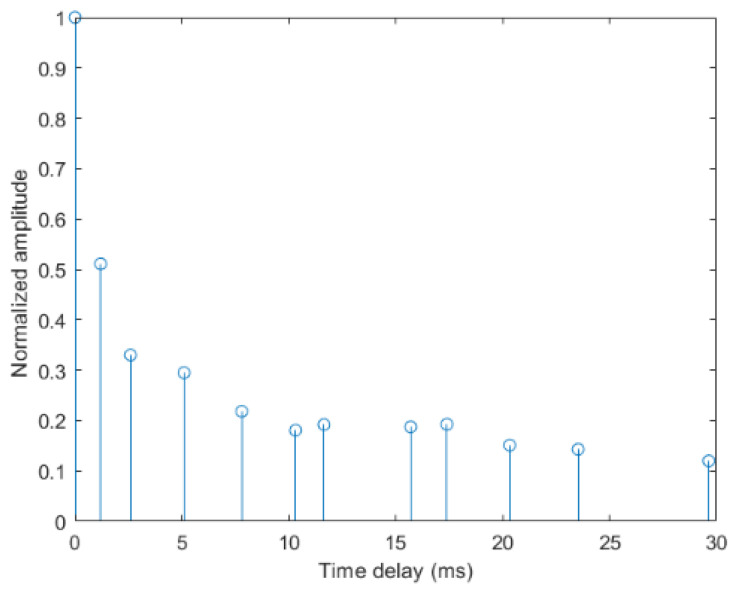
The impulse response of measured underwater channel.

**Figure 15 entropy-24-00720-f015:**
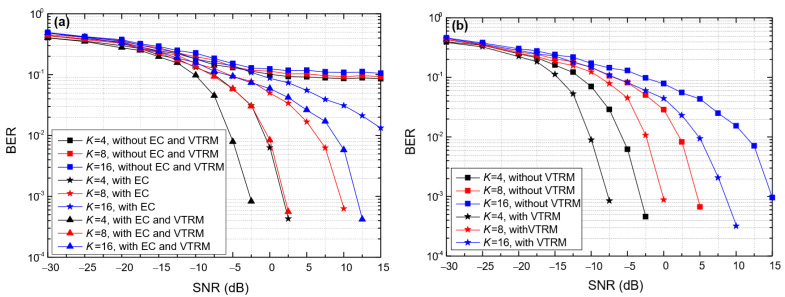
The impact of *T*_sym_ on the BER of the proposed BC-UAC method under the measured channel. (**a**) BC-UAC-TD; (**b**) BC-UAC-FS.

**Figure 16 entropy-24-00720-f016:**
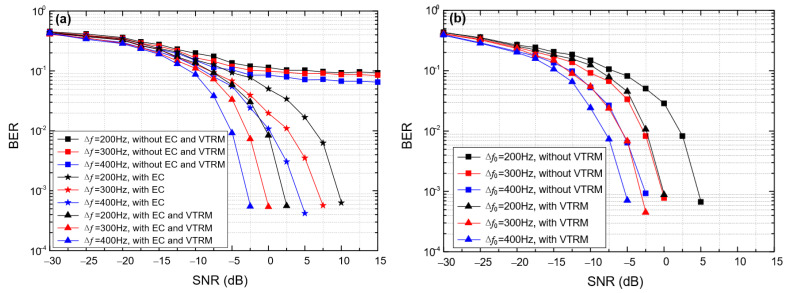
The impact of the frequency offset on the BER of the proposed BC-UAC method under the measured channel. (**a**) BC-UAC-TD (∆*f*); (**b**) BC-UAC-FS (∆*f*_0_).

**Figure 17 entropy-24-00720-f017:**
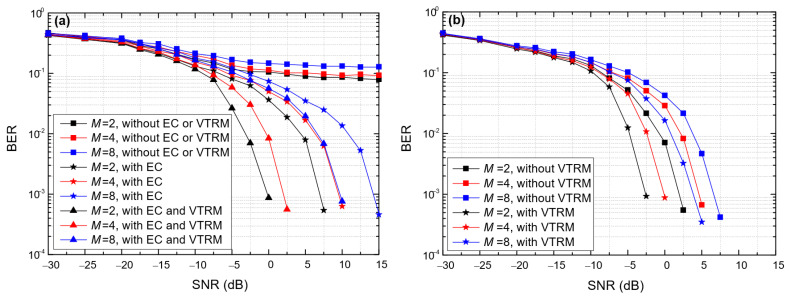
The impact of *M* on the BER of proposed BC-UAC method under the measured channel. (**a**) BC-UAC-TD; (**b**) BC-UAC-FS.

**Table 1 entropy-24-00720-t001:** The impact of *T*_sym_ on the PCC.

Segment Number	Duration of a Symbol (s)	Transmission Rate (bps)	PCC	
			BC-UAC-TD	BC-UAC-FS
4	0.132	15.09	0.99745	0.98919
8	0.066	30.19	0.99744	0.99019
16	0.033	60.38	0.99744	0.99232

**Table 2 entropy-24-00720-t002:** The impact of ∆*f/*∆*f*_0_ on the PCC.

∆*f*/∆*f*_0_ (Hz)	Transmission Rate (bps)	PCC	
		BC-UAC-TD	BC-UAC-FS
200	30.19	0.99744	0.99019
300	30.19	0.99569	0.97730
400	30.19	0.99236	0.95881

**Table 3 entropy-24-00720-t003:** The impact of *M* on the PCC.

Modulation Order, *M*	Transmission Rate (bps)	PCC	
		BC-UAC-TD	BC-UAC-FS
2	15.09	0.99743	0.99770
4	30.19	0.99744	0.99019
8	45.28	0.99888	0.97385

**Table 4 entropy-24-00720-t004:** BELLHOP parameters.

Parameter	Value	Parameter	Value
Distance	5 km	Depth	50 m
Sound speed	1543–1546 m/s	Wave height	0.6 m
Seawater acoustic absorption coefficient	6.94 × 10^−5^ dB/km	Seafloor acoustic absorption coefficient	0.5 dB/km
Seafloor density	1810 kg/m^3^	Seawater density	1021 kg/m^3^
Transmitter transducer depth	25 m	Receiver transducer depth	25 m
Number of beams	10	Transducer beam angle	0~30°

## Data Availability

Not applicable.

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
