# Peer review of "Bionic Covert Underwater Acoustic Communication Based on Time–Frequency Contour of Bottlenose Dolphin Whistle"

_entropy, 2022, doi:10.3390/e24050720_

Round 1

Reviewer 1 Report

The authors have some interesting results and analyses. They understand this topic, related literature and methodologies well.  But for the methods, some improvements are necessary. This includes the use of modern methods such Federated Learning (FL), which can do a lot of work for what authors are doing. FL in 6G can do proper and better jobs. Please find a few references, eg.

An Improved Federated Learning Algorithm for Privacy-Preserving in Cybertwin-Driven 6G System. IEEE Transactions on Industrial Informatics (2022).

Please describe some new technologies that can perform similar/better tasks, or resolve some problems, such as FL in 6G and cite a few references.

Please perform a thorough proofreading.

Reviewer 2 Report

The paper presented for review is interesting, however, the discussion should be further elaborated. The study was well-designed, time-consuming and laborious. I would recommend this paper to be accepted after revising carefully. However, there were a lot of English grammatical mistakes. The introduction of this study are quite short. The authors should overview more about mathematical model. They should elaborate why this research is necessary and how it contribute to practical application. In discussion, the results of this article were compared to previous findings.

Reviewer 3 Report

The manuscript has been improved with respect to its previous version. 

I think that only an English revision is needed now.

Round 2

Reviewer 2 Report

After the changes made, from my point of view, the article can be published as it is.